# Deep Architecture Connectivity Matters for Its Convergence: A Fine-Grained Analysis

**Wuyang Chen**[*]
University of Texas at Austin

**Wei Huang**[*]
RIKEN AIP

**Xinyu Gong**
University of Texas at Austin

**Boris Hanin**
Princeton University

**Zhangyang Wang**
University of Texas at Austin

## Abstract

Advanced deep neural networks (DNNs), designed by either human or AutoML algorithms, are growing increasingly complex. Diverse operations are connected by complicated connectivity patterns, e.g., various types of skip connections. Those topological compositions are empirically effective and observed to smooth the loss landscape and facilitate the gradient flow in general. However, it remains elusive to derive any principled understanding of their effects on the DNN capacity or trainability, and to understand why or in which aspect one specific connectivity pattern is better than another. In this work, we theoretically characterize the impact of connectivity patterns on the convergence of DNNs under gradient descent training in fine granularity. By analyzing a wide network's Neural Network Gaussian Process (NNGP), we are able to depict how the spectrum of an NNGP kernel propagates through a particular connectivity pattern, and how that affects the bound of convergence rates. As one practical implication of our results, we show that by a simple filtration on "unpromising" connectivity patterns, we can trim down the number of models to evaluate, and significantly accelerate the large-scale neural architecture search without any overhead. Code is available at: https://github.com/VITA-Group/architecture_convergence.

## 1 Introduction

Recent years have witnessed substantial progress in designing better deep neural network architectures. The common objective is to build networks that are easy to optimize, of superior trade-offs between efficiency and accuracy, and are generalizable to diverse tasks and datasets. When developing deep networks, how operations (linear transformations and non-linear activations) are connected and stacked together is vital, which is studied in network's convergence [18, 71, 75], complexity [47, 50, 21], generalization [14, 8, 61], loss landscapes [36, 20, 51], etc.

Although it has been widely observed that the performance of deep networks keeps being improved by advanced design options, our understanding remains limited on how a network's properties are influenced by its architectures. For example, in computer vision, design trends have shifted from vanilla chain-like stacked layers [33, 32, 53] to manually elaborated connectivity patterns (ResNet [25], DenseNet [28], etc.). While people observed smoothed loss landscapes [36], mitigated gradient vanishing problem [4], and better generalization [29], these findings only explain the effectiveness of adding skip connections in general, but barely lead to further "finer-grained" insight on more sophisticated composition of skip connections beyond ResNet. Recently, the AutoML community tries to relieve human efforts and propose to automatically discover novel networks from

---

[*]Equal Contribution.

36th Conference on Neural Information Processing Systems (NeurIPS 2022).

gigantic architecture spaces [72, 46]. Despite the strong performance [57, 27, 62, 58], the searched architectures are often composed of highly complex (or even randomly wired) connections, leaving it challenging to analyze theoretically.

Understanding the principles of deep architecture connectivity is of significant importance. Scientifically, this helps answer why composing complicated skip connection patterns has been such an effective "trick" in improving deep networks' empirical performance. Practically, this has direct guidance in designing more efficient and expressive architectures. To close the gap between our theoretical understandings and practical architectures, in this work we target two concrete questions:

**Q1:** Can we understand the precise roles of different connectivity patterns in deep networks?

**Q2:** Can we summarize principles on how connectivity should be designed in deep networks?

One standard way to study how operations and connections affect the network is to analyze the model's convergence under gradient descent [19, 2, 18]. Here, we systematically study the relationship between the connectivity pattern of a neural network and the bound of its convergence rate. A deep architecture can be viewed as a directed acyclic computational graph (DAG), where feature maps are represented as nodes and operations in different layers are directed edges linking features. Under this formulation, by analyzing the spectrum of the Neural Network Gaussian Process (NNGP) kernel, we show that the bound of the convergence rate of the DAG is jointly determined by the number of unique paths in a DAG and the number of parameterized operations on each path. Note that, although several prior arts [44, 1, 10, 11] explored deep learning theories to predict the promise of architectures, their indicators were developed for the general deep network *functions*, not for fine-grained

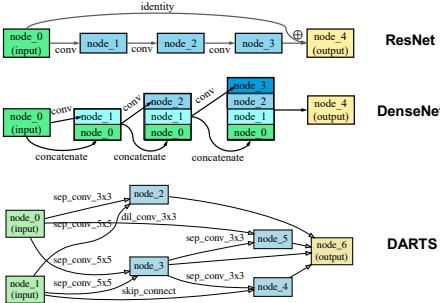

Figure 1: Connectivity patterns of deep networks are growingly more complex, including ResNet [25], DenseNet [28], and architectures sampled from the DARTS search space [40] commonly used in neural architecture search (NAS).

characterization for specific deep architecture *topology patterns*. Therefore, their correlations for selecting better architecture topologies are only *empirically observed*, but not *theoretically justified*. In contrast, our fine-grained conclusion is theory-ground and also empirically verified.

Based on this conclusion, we present two intuitive and practical principles for designing deep architecture DAGs: the "effective depth" $\bar{d}$ and the "effective width" $\bar{m}$. Experiments on diverse architecture benchmarks and datasets demonstrate that $\bar{d}$ and $\bar{m}$ can jointly distinguish promising connectivity patterns from unpromising ones. As a practical implication, our work also suggests a cheap "plug-and-play" method to accelerate the neural architecture search by filtering out potentially unpromising architectures at almost zero cost, before any gradient-based architecture evaluations. Our contributions are summarized below:

- We first theoretically analyze the convergence of gradient descent of diverse neural network architectures, and find the connectivity patterns largely impact their bound of convergence rate.

- From the theoretical analysis, we abstract two practical principles on designing the network's connectivity pattern: "effective depth" $\bar{d}$ and "effective width" $\bar{m}$.

- Both our convergence analysis and principles on effective depth/width are verified by experiments on diverse architectures and datasets. Our method can further significantly accelerate the neural architecture search without introducing any extra cost.

## 2 Related works

### 2.1 Global convergence of deep networks

Many works analyzed the convergence of networks trained with gradient descent [31, 68, 22, 9]. Convergence rates were originally explored for wide two-layer neural networks [60, 54, 55, 38, 19, 43, 3]. More recent works extended the analysis to deeper neural networks [2, 18, 42, 74, 71, 75, 30] and showed that over-parameterized networks can converge to the global minimum with random initialization if the width (number of neurons) is polynomially large as the number of training samples. In comparison, we expand such analysis to bridge the gap between theoretical understandings of general neural networks and practical selections of better "fine-grained" architectures.

## 2.2 Residual structures in deep networks

Architectures for computer vision have evolved from plain chain-like stacked layers [33, 32, 53] to elaborated connectivity patterns (ResNet [25], DenseNet [28], etc.). With the development of AutoML algorithms, novel networks of complicated operations/connections were discovered. Despite their strong performance [57, 27, 62, 58], these architectures are often composed of highly complex connections and are hard to analyze. [52] defined the depth and width for complex connectivity patterns, and studied the impacts of network topologies in different cases. To better understand these residual patterns, people tried to unify the architectures with different formulations. One seminal way is to represent network structures into graphs, then randomly sample different architectures from the graph distribution and empirically correlate their generalization with graph-related metrics [62, 67]. Other possible ways include geometric topology analysis [6], network science [5], etc. For example, [5] proposed NN-Mass by analyzing the network's layer-wise dynamic isometry, and then empirically linked to the network's convergence. However, none of those works directly connect the convergence rate analysis to different residual structures.

## 2.3 Theory-guided design of neural architectures

Particularly related to our work is an emerging research direction, that tries to connect recent deep learning theories to guide the design of novel network architectures. The main idea is to find theoretical indicators that have strong correlations with network's training or testing performance. [44] pioneered a training-free NAS method, which empirically leveraged sample-wise activation patterns to rank architectures. [45] leveraged the network's NNGP features to approximate its predictions. Different training-free indicators were evaluated in [1], and the "synflow" measure [59] was leveraged as the main ranking metric. [10] incorporated two theory-inspired metrics with supernet pruning as the search method. However, these works mainly adapted theoretical properties of the general deep neural network *function*: their correlations with the concrete network architecture *topology* are only *empirically observed*, but not *theoretically justified*.

## 3 Topology matters: convergence analysis with connectivity patterns

In this section, we study the convergence of gradient descent for deep networks, whose connectivity patterns can be formulated as small but representative direct acyclic graphs (DAGs). Our goal is to compare the convergence rate bounds of different DAGs, and further establish links to their connectivity patterns, leading to abstracting design principles.

### 3.1 Problem setup and architectures notations

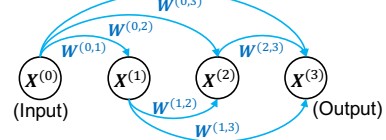

We consider a network's DAG as illustrated in Figure 2. $\boldsymbol{X}^{(h)}$ ($h \in [0, H]$) is the feature map (node) and $\boldsymbol{W}$ is the operation (edge). $\boldsymbol{X}^{(0)}$ is the input node, $\boldsymbol{X}^{(H)}$ is the output node ($H = 3$ in this case), and $\boldsymbol{X}^{(1)}, \cdots, \boldsymbol{X}^{(H-1)}$ are intermediate nodes. The DAG is allowed to be fully connected: any two nodes could be connected by an operation. Feature maps from multiple edges coming to one vertex will be directly summed up. This DAG has many practical instances: for example, it is used in NAS-Bench-201 [17], a popular NAS benchmark. The forward process of the network in Figure 2 can be formulated as:

Figure 2: A network represented as a direct acyclic graph (DAG). $\boldsymbol{X}^{(h)}$ is the feature map (node). $\boldsymbol{W}$ is the operation (edge). $h \in [0, H]$, here $H = 3$. If we remove some edges or set some $\boldsymbol{W}$ as skip-connections (i.e., identity mappings), how would the convergence of this DAG be affected?

$$
\begin{aligned}
\boldsymbol{X}^{(1)} &= \rho(\boldsymbol{W}^{(0,1)}\boldsymbol{X}^{(0)}) \\
\boldsymbol{X}^{(2)} &= \rho(\boldsymbol{W}^{(0,2)}\boldsymbol{X}^{(0)}) + \rho(\boldsymbol{W}^{(1,2)}\boldsymbol{X}^{(1)}) \\
\boldsymbol{X}^{(3)} &= \rho(\boldsymbol{W}^{(0,3)}\boldsymbol{X}^{(0)}) + \rho(\boldsymbol{W}^{(1,3)}\boldsymbol{X}^{(1)}) + \rho(\boldsymbol{W}^{(2,3)}\boldsymbol{X}^{(2)}) \\
\boldsymbol{u} &= \boldsymbol{a}^\top \boldsymbol{X}^{(3)}.
\end{aligned} \tag{1}
$$

Feature $\boldsymbol{X}^{(s)} \in \mathbb{R}^{m \times 1}$, where $m$ is the absolute width of an edge (i.e. number of neurons), and $s \in \{1, 2, 3\}$. $\boldsymbol{a}$ is the final layer and $\boldsymbol{u}$ is the network's output. We consider three candidate operations for each edge: a linear transformation followed by a non-linear activation, or a skip-

connection (identity mapping), or a broken edge (zero mapping):

$$\boldsymbol{W}^{(s,t)}\begin{cases} = \boldsymbol{0} & \text{zero} \\ = \boldsymbol{I}^{m\times m} & \text{skip-connection} \\ \sim \mathcal{N}(\boldsymbol{0}, \boldsymbol{I}^{m\times m}) & \text{linear transformation} \end{cases}, \quad \rho(x) = \begin{cases} 0 & \text{zero} \\ 1 & \text{skip-connection} \\ \sqrt{\frac{c_\sigma}{m}}\sigma(x), & \text{linear transformation} \end{cases} . \quad (2)$$

$s,t \in \{1,2,3\}$, $\mathcal{N}$ is the Gaussian distribution, and $\sigma$ is the activation function. $c_\sigma = \left(\mathbb{E}_{x\sim N(0,1)}\left[\sigma(x)^2\right]\right)^{-1}$ is a scaling factor to normalize the input in the initialization phase.

Consider the Neural Network Gaussian Process (NNGP) in the infinite-width limit [34], we define our NNGP variance as $\boldsymbol{K}_{ij}^{(s)} = \langle \boldsymbol{X}_i^{(s)}, \boldsymbol{X}_j^{(s)} \rangle$ and NNGP mean as $\boldsymbol{b}_i^{(s)} = \mathbb{E}[\boldsymbol{X}_i^{(s)}]$, $i,j \in 1,\cdots,N$ for $N$ training samples in total. Both $\boldsymbol{K}_{ij}$ and $\boldsymbol{b}_i$ are taken the expectation over the weight distributions.

## 3.2 Preliminary: bounding the network's linear convergence rate via NNGP spectrum

Before we analyze different connectivity patterns (Section 3.3), we first give the linear convergence of our DAG networks (Theorem 3.1) and also show the guarantee of the full-rankness of $\lambda_{\min}(\boldsymbol{K}^{(H)})$ (Lemma 3.1). For a sufficiently wide neural network, its bound of convergence rate to the global minimum can be governed by the NNGP kernel. The linear convergence rate for a deep neural network of a DAG-like connectivity pattern is shown as follows:

**Theorem 3.1** (Linear Convergence of DAG). *Consider a DAG of $H$ nodes and $P_H$ end-to-end paths. At $k$-th gradient descent step on $N$ training samples, with MSE loss $\mathcal{L}(k) = \frac{1}{2}\|\boldsymbol{y}-\boldsymbol{u}(k)\|_2^2$, suppose the learning rate $\eta = O\left(\frac{\lambda_{\min}(\boldsymbol{K}^{(H)})}{(NP_H)^2}2^{O(H)}\right)$ and the number of neurons per layer $m = \Omega\left(\max\left\{\frac{(NP_H)^4}{\lambda_{\min}^4(\boldsymbol{K}^{(H)})}, \frac{NP_H H}{\delta}, \frac{(NP_H)^2 \log\left(\frac{HN}{\delta}\right)2^{O(H)}}{\lambda_{\min}^2(\boldsymbol{K}^{(H)})}\right\}\right)$, we have*

$$\|\boldsymbol{y}-\boldsymbol{u}(k)\|_2^2 \le \left(1 - \frac{\eta\lambda_{\min}(\boldsymbol{K}^{(H)})}{2}\right)^k \|\boldsymbol{y}-\boldsymbol{u}(0)\|_2^2, \quad (3)$$

*where $P_H$ is number of end-to-end paths from $\boldsymbol{X}^{(0)}$ to $\boldsymbol{X}^{(H)}$.*

*Remark* 3.2. In our work, we will use a fixed small learning rate $\eta$ across different network architectures, to focus our analysis on the impact of $\lambda_{\min}(\boldsymbol{K}^{(H)})$. This is motivated by the widely adopted setting in popular architecture benchmarks [48, 17] that we will follow in our experiments.

The above theorem shows that the convergence rate is bounded by the smallest eigenvalue of $\boldsymbol{K}^{(H)}$, and also indicates that networks of larger least eigenvalues will more likely to converge faster. This requires that the $\boldsymbol{K}^{(H)}$ has full rankness, which is demonstrated by the following lemma:

**Lemma 3.1** (Full Rankness of $\boldsymbol{K}^{(H)}$). *Suppose $\sigma(\cdot)$ is analytic and not a polynomial function. If no parallel data points, then $\lambda_{\min}\left(\boldsymbol{K}^{(H)}\right) > 0$.*

## 3.3 How does NNGP propagate through DAG?

Now we are ready to link the DAG's connectivity pattern to the bound of its convergence rate. Although different DAGs are all of the linear convergence under gradient descent, they are very likely to have different bounds of convergence rates. Finding the exact mapping from $\lambda_{\min}(\boldsymbol{K}^{(0)})$ to $\lambda_{\min}(\boldsymbol{K}^{(H)})$ will lead us to a fine-grained comparison between different connectivity patterns.

First, for fully-connected operations, we can obtain the propagation of the NNGP variance and mean between two consecutive layers:

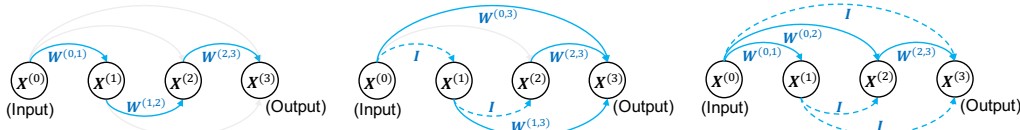

Figure 3: Three example DAGs. Solid blue arrows are parameterized operations (e.g. fully-connected layer). Dashed arrows are non-parameterized operations (e.g. skip-connections). Removed edges are in light grey. See Section 3.3 for their convergence analysis. Left: sequential connectivity (DAG#1). Middle: parallel connectivity (DAG#2). Right: mixed of sequential and parallel connectivity (DAG#3).

**Lemma 3.2** (Propagation of $\boldsymbol{K}$). *Define the propagation as $\boldsymbol{K}^{(l)} = f(\boldsymbol{K}^{(l-1)})$ and $\boldsymbol{b}^{(l)} = g(\boldsymbol{b}^{(l-1)})$. When the edge operation is a linear transformation, we have:*

$$\boldsymbol{K}_{ii}^{(l)} = f(\boldsymbol{K}_{ii}^{(l-1)}) = \int \mathcal{D}_z c_\sigma \sigma^2\left(\sqrt{\boldsymbol{K}_{ii}^{(l-1)}}z\right)$$

$$\boldsymbol{K}_{ij}^{(l)} = f(\boldsymbol{K}_{ij}^{(l-1)}) = \int \mathcal{D}_{z_1}\mathcal{D}_{z_2} c_\sigma \sigma\left(\sqrt{\boldsymbol{K}_{ii}^{(l-1)}}z_1\right)\sigma\left(\sqrt{\boldsymbol{K}_{jj}^{(l-1)}}(\boldsymbol{C}_{ij}^{(l-1)}z_1 + \sqrt{1 - (\boldsymbol{C}_{ij}^{(l-1)})^2}z_2)\right)$$

$$\boldsymbol{C}_{ij}^{(l)} = \boldsymbol{K}_{ij}^{(l)}/\sqrt{\boldsymbol{K}_{ii}^{(l)}\boldsymbol{K}_{jj}^{(l)}}$$

$$\boldsymbol{b}_{i}^{(l)} = g(\boldsymbol{b}_{i}^{(l-1)}) = \int \mathcal{D}_z \sqrt{c_\sigma}\,\sigma\left(\sqrt{\boldsymbol{K}_{ii}^{(l)}}z\right)$$

(4)

*where $z$, $z_1$ and $z_2$ are independent standard Gaussian random variables. Besides, $\int \mathcal{D}_z = \frac{1}{\sqrt{2\pi}}\int dz e^{-\frac{1}{2}z^2}$ is the measure for a normal distribution.*

*Remark* 3.3. Since $\boldsymbol{b}_i^{(l)}$ is a constant, it will not affect our analysis and we will omit it below.

*Remark* 3.4. With centered normalized inputs, $\boldsymbol{K}_{ii}^{(0)} = 1$.

Our last lemma to prepare states that we can bound the smallest eigenvalue of the NNGP kernel of $N \times N$ by its $2 \times 2$ principal submatrix case.

**Lemma 3.3.** *For a positive definite symmetric matrix $\boldsymbol{K} \in \mathbb{R}^{N \times N}$, the smallest eigenvalue is bounded by the smallest eigenvalue of its $2 \times 2$ principal sub-matrix.*

$$\lambda_{\min}(\boldsymbol{K}) \leq \min_{i \neq j} \lambda_{\min}\begin{bmatrix} \boldsymbol{K}_{ii} & \boldsymbol{K}_{ij} \\ \boldsymbol{K}_{ji} & \boldsymbol{K}_{jj} \end{bmatrix}$$

(5)

Now we can analyze the smallest eigenvalue $\lambda_{\min}(\boldsymbol{K}^{(H)})$ for different DAGs. Note that here we adopt ReLU activation for three reasons: 1) Analyzing ReLU with the initialization at the edge-of-chaos [23] exhibits promising results; 2) Previous works also imply that the convergence rate of ReLU-based networks depends on $\lambda_0$ [19]; 3) ReLU is the most commonly used activation in practice. Specifically, we set $c_\sigma = 2$ for ReLU [18, 23]. In Figure 3, we show three representative DAGs ($H = 3$). For a fair comparison, each DAG has three linear transformation layers, i.e., they have the same number of parameters and mainly differ in how nodes are connected. After these three case studies, we will show a more general rule for the propagation of $\boldsymbol{K}^{(0)}$.

**DAG#1 (sequential connections).** In this case, $\boldsymbol{W}^{(0,1)}, \boldsymbol{W}^{(1,2)}, \boldsymbol{W}^{(2,3)} =$ linear transformation, and $\boldsymbol{W}^{(0,2)}, \boldsymbol{W}^{(0,3)}, \boldsymbol{W}^{(1,2)} =$ zero (broken edge). There is only one unique path connecting from $\boldsymbol{X}^{(0)}$ to $\boldsymbol{X}^{(3)}$, with three parameterized operations on it.

$$\boldsymbol{K}^{(1)} = f(\boldsymbol{K}^{(0)}) \qquad \boldsymbol{K}^{(2)} = f(\boldsymbol{K}^{(1)}) \qquad \boldsymbol{K}^{(3)} = f(\boldsymbol{K}^{(2)})$$

(6)

By Lemma 3.3, we calculate the upper bound of the least eigenvalue of $\boldsymbol{K}^{(3)}$, denoted as $\lambda_{\text{dag1}}$:

$$\lambda_{\min}\begin{bmatrix} \boldsymbol{K}_{ii}^{(3)} & \boldsymbol{K}_{ij}^{(3)} \\ \boldsymbol{K}_{ji}^{(3)} & \boldsymbol{K}_{jj}^{(3)} \end{bmatrix} = \lambda_{\min}\begin{bmatrix} \boldsymbol{K}_{ii}^{(2)} & f(\boldsymbol{K}_{ij}^{(2)}) \\ f(\boldsymbol{K}_{ji}^{(2)}) & \boldsymbol{K}_{jj}^{(2)} \end{bmatrix} = \lambda_{\min}\begin{bmatrix} \boldsymbol{K}_{ii}^{(1)} & f(f(\boldsymbol{K}_{ij}^{(1)})) \\ f(f(\boldsymbol{K}_{ji}^{(1)})) & \boldsymbol{K}_{jj}^{(1)} \end{bmatrix}$$

$$= \lambda_{\min}\begin{bmatrix} \boldsymbol{K}_{ii}^{(0)} & f(f(f(\boldsymbol{K}_{ij}^{(0)}))) \\ f(f(f(\boldsymbol{K}_{j,i}^{(0)}))) & \boldsymbol{K}_{jj}^{(0)} \end{bmatrix} = 1 - f^3(\boldsymbol{K}_{ij}^{(0)}) \equiv \lambda_{\text{dag1}},$$

(7)

where we denote the function composition $f^3(\boldsymbol{K}_{ij}^{(0)}) = f(f(f(\boldsymbol{K}_{ij}^{(0)})))$.

**DAG#2 (parallel connections).** In this case, $\boldsymbol{W}^{(0,3)}, \boldsymbol{W}^{(1,3)}, \boldsymbol{W}^{(2,3)}$ = linear transformation, $\boldsymbol{W}^{(0,1)}, \boldsymbol{W}^{(1,2)}$ = skip-connection, and $\boldsymbol{W}^{(0,2)}$ = zero (broken edge). There are three unique paths connecting from $\boldsymbol{X}^{(0)}$ to $\boldsymbol{X}^{(3)}$, with one parameterized operation on each path.

$$\boldsymbol{K}^{(2)} = \boldsymbol{K}^{(1)} = \boldsymbol{K}^{(0)}$$
$$\boldsymbol{K}^{(3)} = f(\boldsymbol{K}^{(0)}) + f(\boldsymbol{K}^{(1)}) + f(\boldsymbol{K}^{(2)}) = 3f(\boldsymbol{K}^{(0)}) \tag{8}$$

where $\boldsymbol{C}$ is a constant matrix with all entries being the same. Therefore:

$$
\lambda_{\min}
\begin{bmatrix}
\boldsymbol{K}_{ii}^{(3)} & \boldsymbol{K}_{ij}^{(3)} \\
\boldsymbol{K}_{ji}^{(3)} & \boldsymbol{K}_{jj}^{(3)}
\end{bmatrix}
= \lambda_{\min}
\begin{bmatrix}
3\boldsymbol{K}_{ii}^{(0)} & 3f(\boldsymbol{K}_{ij}^{(0)}) \\
3f(\boldsymbol{K}_{ji}^{(0)}) & 3\boldsymbol{K}_{jj}^{(0)}
\end{bmatrix}
$$
$$
= 3\lambda_{\min}
\begin{bmatrix}
\boldsymbol{K}_{ii}^{(0)} & f(\boldsymbol{K}_{ij}^{(0)}) \\
f(\boldsymbol{K}_{ji}^{(0)}) & \boldsymbol{K}_{jj}^{(0)}
\end{bmatrix}
= 3(1 - f(\boldsymbol{K}_{ij}^{(0)})) \equiv \lambda_{\text{dag2}}. \tag{9}
$$

**DAG#3 (mixture of sequential and parallel connections).** In this case, $\boldsymbol{W}^{(0,1)}, \boldsymbol{W}^{(0,2)}, \boldsymbol{W}^{(2,3)}$ = linear transformation, $\boldsymbol{W}^{(0,3)}, \boldsymbol{W}^{(1,2)}, \boldsymbol{W}^{(1,3)}$ = skip connection. There are four unique paths connecting from $\boldsymbol{X}^{(0)}$ to $\boldsymbol{X}^{(3)}$, with 0/1/2 parameterized operations on each path.

$$\boldsymbol{K}^{(1)} = f(\boldsymbol{K}^{(0)}) \qquad \boldsymbol{K}^{(2)} = \boldsymbol{K}^{(1)} + f(\boldsymbol{K}^{(0)})$$
$$\boldsymbol{K}^{(3)} = f(\boldsymbol{K}^{(2)}) + \boldsymbol{K}^{(1)} + \boldsymbol{K}^{(0)} = \boldsymbol{K}^{(0)} + f(\boldsymbol{K}^{(0)}) + f(2f(\boldsymbol{K}^{(0)})) \tag{10}$$

Then we have:

$$
\lambda_{\min}
\begin{bmatrix}
\boldsymbol{K}_{ii}^{(3)} & \boldsymbol{K}_{ij}^{(3)} \\
\boldsymbol{K}_{j,i}^{(3)} & \boldsymbol{K}_{jj}^{(3)}
\end{bmatrix}
= \lambda_{\min}
\begin{bmatrix}
4\boldsymbol{K}_{ii}^{(0)} & \tilde{f}(\boldsymbol{K}_{ij}^{(0)}) \\
\tilde{f}(\boldsymbol{K}_{j,i}^{(0)}) & 4\boldsymbol{K}_{jj}^{(0)}
\end{bmatrix}
= 4 - \tilde{f}(\boldsymbol{K}_{j,i}^{(0)}) \equiv \lambda_{\text{dag3}}, \tag{11}
$$

where $\tilde{f}(\boldsymbol{K}_{ij}^{(0)}) = \boldsymbol{K}_{ij}^{(0)} + f(\boldsymbol{K}_{ij}^{(0)}) + f(2f(\boldsymbol{K}_{ij}^{(0)}))$

**Conclusion:** by comparing Eq. 7, 9, and 11, we can show the rank of three upper bounds of least eigenvalues as: $\lambda_{\text{dag1}} < \lambda_{\text{dag2}} < \lambda_{\text{dag3}}$. Therefore, the bound of the convergence rate of DAG#3 is better than DAG#2, and DAG#1 is the worst. See our Appendix C in supplement for detailed analysis.

**General rules of propagation from $\boldsymbol{K}^{(0)}$ to $\boldsymbol{K}^{(H)}$.**

- Diagonal elements of $\boldsymbol{K}^{(H)}$ is determined by the number of unique paths from $\boldsymbol{X}^{(0)}$ to $\boldsymbol{X}^{(H)}$.

- Off-diagonal elements of $\boldsymbol{K}^{(H)}$ is determined by the number of incoming edges of $\boldsymbol{X}^{(H)}$ and the number of parameterized operations on each path.

Thus, we could simplify our rules as:

$$\lambda_{\min}(\boldsymbol{K}^{(H)}) \leq \min_{i \neq j} \left( P - \sum_{p=1}^{P} f^{d_p}\left(\boldsymbol{K}_{ij}^{(0)}\right) \right), \tag{12}$$

where $P$ is number of end-to-end paths from $\boldsymbol{X}^{(0)}$ to $\boldsymbol{X}^{(H)}$, $d_p$ is the number of linear transformation operations on the $p$-th path, and $f^{d_p}$ indicates a $d_p$-power composition of $f$. This summary is based on the propagation of the NNGP variance. With this rule, we can quickly link the relationship between the smallest eigenvalue of $\boldsymbol{K}^{(H)}$ and $\boldsymbol{K}^{(0)}$.

## 4 Practical principle of connectivity: effective depth and effective width

Eq. 12 precisely characterizes the impact of a network's topology on its convergence. Inspired by above results, we observe two factors that control the $\lambda_{\min}(\boldsymbol{K}^{(H)})$: (1) $P$, the "width" of a DAG; (2) $d_p(p \in [1, P])$, the "depth" of a DAG. However, directly comparing convergences via Eq. 12 is non-trivial (see our Appendix C), because: (1) the complicated form of the NNGP propagation "$f$" (Eq. 3.2); (2) "$P$" and "$d_p$"s are not truly free variables, as they are still coupled together in the discrete topology space of a fully connected DAG.

Motivated by these two challenges, we propose a practical version of our principle. We simplify the analysis of Eq. 12 by reducing "$P$" and "$d_p$"s to only two intuitive variables highly related to the network's connectivity patterns: the effective depth and effective width.

**Definition 4.1** (Effective Depth and Width). Consider the directed acyclic graph (DAG) of a network. Suppose there are $P$ unique paths connecting the starting and the ending vertex. Denote the number of parameterized operations on the $p$-th path as $d_p, p \in [1, P]$. We define:

$$\text{Effective Depth} \quad \bar{d} = \frac{\sum_{p=1}^{P} d_p}{P}, \qquad \text{Effective Width} \quad \bar{m} = \frac{\sum_{p=1}^{P} \mathbb{1}(d_p > 0)}{\bar{d}}, \qquad (13)$$

where $\mathbb{1}(d_p > 0) = 1$ if $d_p > 0$ otherwise is $0$.

*Remark* 4.2. In our experiments below, we consider fully-connected or convolutional layers as parameterized operations, ignoring their detailed configurations (kernel sizes, dilations, groups, etc.). We consider skip-connections and pooling layers as non-parameterized operations.

The effective depth considers the averaging effects of multiple parallel paths, and the effective width considers the amount of unique information flowing from the starting vertex to the end, normalized by the overall depth. We will demonstrate later in Section 5.2 that performances of networks from diverse architecture spaces show positive correlations with these two aspects. While $\bar{d}$ and $\bar{m}$ may be loose to predict the best architecture, in Section 5.3 we will show that they are very stable in distinguishing bad ones. Therefore, using these two principles, we can quickly determine if an architecture is potentially unpromising at almost zero cost, by only analyzing its connectivity without any forward or backward calculations.

## 5 Experiments

### 5.1 Empirical convergence confirms our analysis

We first experimentally verify our convergence analysis in Section 3.3. In all cases we use ReLU nonlinearities with Kaiming normal initialization [24]. We build the same three computational graphs of fully-connected layers in Figure 3. Three networks have hidden layers of a constant width of 1024. We train the network using SGD with a mini-batch of size 128. The learning rate is fixed at $1 \times 10^{-5}$. No augmentation, weight decay, learning rate decay, or momentum is adopted.

Based on our analysis, on both MNIST and CIFAR-10, the convergence rate of DAG#1 (Figure 3 left) is worse than DAG#2 (Figure 3 middle), and is further worse than DAG#3 (Figure 3 right). The experimental observation is consistent with this analysis: the training accuracy of DAG#3 increases the fastest, and DAG#2 is faster than DAG#1. Although on CIFAR-10 DAG#1 slightly outperforms DAG#2 in the end, DAG#2 still benefits from

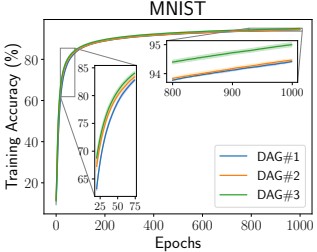

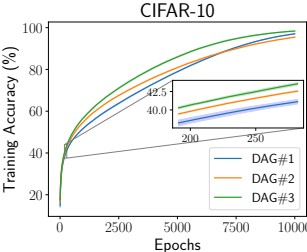

Figure 4: Empirical convergence of three DAG cases (Figure 3) can verify our theoretical analysis in Section 3. Small standard deviations of three runs are shown in shadows.

faster convergence in most early epochs. This result confirms that by our convergence analysis, we can effectively compare the convergence of different connectivity patterns.

### 5.2 Extreme $\bar{d}$ or $\bar{m}$ leads to bad performance on architecture benchmarks

In this experiment, our goal is to verify the two principles we proposed in Section 4. Specifically, we will leverage a large number of network architectures of random connectivities, calculate their $\bar{d}$ and $\bar{m}$, and compare their performance. We consider popular architecture benchmarks that are widely studied in the community of neural architecture search (NAS). Licenses are publicly available.

- The *NAS-Bench-201* [17] provides 15,625 architectures that are stacked by repeated DAGs of four nodes (exactly the same DAG we considered in Section 3 and Figure 2). It contains architecture's performance on three datasets (CIFAR-10, CIFAR-100, ImageNet-16-120 [15]) evaluated under a unified protocol (i.e. same learning rate, batch size, etc., for all architectures). The operation space contains *zero*, *skip-connection*, *conv*$1 \times 1$, *conv*$3 \times 3$ *convolution*, and *average pooling* $3 \times 3$.
- Large-scale architecture spaces: NASNet [73], Amoeba [49], PNAS [39], ENAS [46], DARTS [40]. These spaces have more operation types and more complicated rules on allowed DAG connectivity

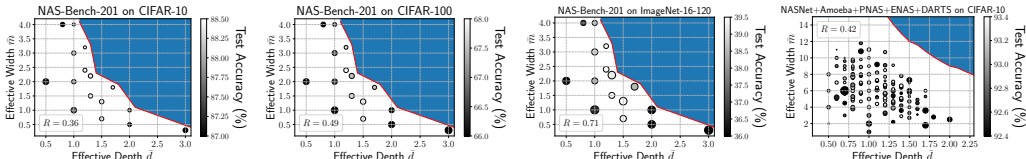

Figure 5: In a complete architecture space, being extremely large or small on either the effective depth ($\bar{d}$) or the effective width ($\bar{m}$) leads a connectivity pattern to bad performance (black) and large variance (large circle size). Each dot represents a subset of architectures of the same $\bar{d}$ and $\bar{m}$. The left three plots are results on CIFAR-10, CIFAR-100, and ImageNet-16-120 from the NAS-Bench-201 [17]. The rightmost plot is on CIFAR-10 from a union of multiple large-scale spaces: NASNet [73], Amoeba [49], PNAS [39], ENAS [46], DARTS [40]. Blue areas indicate invalid DAG regions: a DAG cannot be both deep and wide at the same time.

patterns, see Figure 1 bottom as an example. We refer to their papers for details. A benchmark is further proposed in [48]: about 5000 architectures are randomly selected from each space above, and in total we have 24827 architectures pretrained on CIFAR-10. We will use this prepared data[2].

We show our results in Figure 5. From all four plots[3] (across different architecture spaces and datasets), it can be observed that, architectures of extreme depths or widths suffer from bad performance. We also calculate the multi-correlation[4] ($R$) between the accuracy and the joint of both $\bar{d}$ and $\bar{m}$, and show as legends in Figure 5. All four plots show positive correlations. It is worth noting that here each dot represents a subset of architectures that share the same $\bar{d}$ and $\bar{m}$, with possible different fine-grained topology or layer types, indicating the generalizability of our proposed principles. The variance of performance in each subset is represented by the radius of the dot.

*Remark* 5.1. Our notion of "extreme $\bar{d}$ or $\bar{m}$" targets a complete space of DAGs: given the total number of nodes, any two nodes in the graph can be connected, and there are no topological restrictions or disabled edges (i.e., one cannot introduce any prior to the distribution of graph topologies).

## 5.3  A "Plug-and-Play" method for accelerating NAS: bypassing extreme $\bar{d}$ and $\bar{m}$

Inspired by the bad performance from extreme $\bar{d}$ or $\bar{m}$ discussed in Section 5.2, we are motivated to further contribute a "plug-and-play" tool to accelerate practical NAS applications.

**Method.**  In NAS, the bottleneck of search efficiency is the evaluation cost during the search. Our core idea is to use $\bar{d}$ and $\bar{m}$ to pre-select architectures that are very likely to be unpromising before we spend extra cost to evaluate them. Training-based NAS is slow: one has to perform gradient descent on each sampled architecture and use the loss or accuracy to rank different networks. We instead choose to improve two recent training-free NAS methods: NAS-WOT [44] and TE-NAS [10]. This is because our goal here is to accelerate NAS, and these two are state-of-the-art NAS works of extremely low search time cost. We plan to show that our method can even further accelerate these two training-free NAS works. The rationale is that, our $\bar{d}$ and $\bar{m}$ are even much cheaper: we only do a simple calculation on a network's DAG structure, but NAS-WOT and TE-NAS still have to perform forward or backward on a mini-batch of real data. Although our $\bar{d}$ and $\bar{m}$ are only inspired from the optimization perspective, not the complexity or generalization, our method mainly filters out bad architectures at a coarse level, but does not promote elites in a fine-grained way.

*NAS-WOT*  proposed to rank architectures by their local linear maps. Given a mini-batch of data, each point can be identified by its activation pattern through the network, and a network that can assign more unique activation patterns will be considered promising. NAS-WOT uses random sampling to search the architectures and rank them based on this training-free score. **Our version**: We will skip potentially unpromising architecture before leaving them for NAS-WOT to calculate their scores.

*TE-NAS*  proposed to rank architectures by combining both the trainability (condition number of NTK) and expressivity (number of linear regions). TE-NAS progressively pruned a super network to a single-path network, by removing unpromising edges of low scores.

---

[2]Data is publicly available at `https://github.com/facebookresearch/nds`, only test accuracy available.
[3]To fairly compare different connectivity patterns, we fix the total number of convolutional layers per model as 3 (out of 6 edges) on NAS-Bench-201, and 5 (out of 10 edges) on the union of large-scale architecture spaces.
[4]See Appendix D for its definition.

Table 1: NAS Search Performance in DARTS space on ImageNet. Our standard deviations over three random runs are included in parentheses.

| Architecture | Test Error(%) | | Params. (M) | Search Cost (GPU days) | Search Method |
|---|---|---|---|---|---|
| | top-1 | top-5 | | | |
| NASNet-A [73] | 26.0 | 8.4 | 5.3 | 2000 | RL |
| AmoebaNet-C [49] | 24.3 | 7.6 | 6.4 | 3150 | evolution |
| PNAS [39] | 25.8 | 8.1 | 5.1 | 225 | SMBO |
| MnasNet-92 [56][†] | 25.2 | 8.0 | 4.4 | 288 (TPU) | RL |
| DARTS (2nd) [40] | 26.7 | 8.7 | 4.7 | 4.0[‡] | gradient |
| SNAS (mild) [63] | 27.3 | 9.2 | 4.3 | 1.5 | gradient |
| GDAS [16] | 26.0 | 8.5 | 5.3 | 0.21 | gradient |
| BayesNAS [70] | 26.5 | 8.9 | 3.9 | 0.2 | gradient |
| P-DARTS [13] | 24.4 | 7.4 | 4.9 | 0.3 | gradient |
| PC-DARTS [64] | 25.1 | 7.8 | 5.3 | 0.1[‡] | gradient |
| PC-DARTS (ImageNet) [64][†] | 24.2 | 7.3 | 5.3 | 3.8 | gradient |
| ProxylessNAS (GPU) [7][†] | 24.9 | 7.5 | 7.1 | 8.3 | gradient |
| SGAS (Cri 1. avg) [35] | 24.42 (0.16) | 7.29 (0.09) | 5.3 | 0.25 | gradient |
| DrNAS [12][†] | 23.7 | 7.1 | 5.7 | 4.6 | gradient |
| NAS-WOT [44][†][*] | 26.2 | 8.2 | 4.4 | 0.0036[‡] | training-free |
| NAS-WOT + DAG (ours)[†] | 25.9 (0.4) | 8.2 | 4.4 | 0.003[‡] | training-free |
| TE-NAS [10][†] | 24.5 | 7.5 | 5.4 | 0.17[‡] | training-free |
| TE-NAS + DAG (ours)[†] | 24.2 (0.3) | 7.4 | 6.1 | 0.1[‡] | training-free |

[†] Architectures searched on ImageNet. Other methods searched on CIFAR-10 and then transfered to the ImageNet.
[*] Our reproduced result, the original work did not provide results on the ImageNet.
[‡] Recorded on a single GTX 1080Ti GPU.

Table 2: Search Performance from NAS-Bench-201. "optimal" indicates the best test accuracy achievable in the space. Our standard deviations over three random runs are included in parentheses.

| Architecture | CIFAR-10 | CIFAR-100 | ImageNet-16-120 | Search Cost (GPU sec.) | Search Method |
|---|---|---|---|---|---|
| ResNet [25] | 93.97 | 70.86 | 43.63 | - | - |
| RSPS [37] | 87.66(1.69) | 58.33(4.34) | 31.14(3.88) | 8007.13 | random |
| ENAS [46] | 54.30(0.00) | 15.61(0.00) | 16.32(0.00) | 13314.51 | RL |
| DARTS (1st) [40] | 54.30(0.00) | 15.61(0.00) | 16.32(0.00) | 10889.87 | gradient |
| DARTS (2nd) [40] | 54.30(0.00) | 15.61(0.00) | 16.32(0.00) | 29901.67 | gradient |
| GDAS [16] | 93.61(0.09) | 70.70(0.30) | 41.84(0.90) | 28925.91 | gradient |
| WOT [44] | 92.81 (0.99) | 69.48 (1.70) | 43.10 (3.16) | 30.01 | training-free |
| WOT + DAG (ours) | 92.98 (0.78) | 69.86 (1.24) | 43.44 (2.64) | 17.95 (-40.2%) | training-free |
| TE-NAS [10] | 93.9 (0.47) | 71.24 (0.56) | 42.38 (0.46) | 1558 | training-free |
| TE-NAS + DAG (ours) | 93.6 (0.37) | 71.26 (0.77) | 44.38 (0.76) | 1077 (-30.9%) | training-free |
| FP-NAS [65] | 77.4 (16.6) | 64.7 (5.3) | 26.7 (10.4) | 4837 | gradient |
| FP-NAS + DAG (ours) | 93.3 (0.3) | 70.8 (0.4) | 44.5 (1.4) | 2612 (-46.0%) | gradient |
| **Optimal** | 94.37 | 73.51 | 47.31 | - | - |

> **Our version**: We will skip potentially unpromising architecture before leaving them for TE-NAS to prune.

We provide a pseudocode algorithm in Appendix A to demonstrate the usage of our method.

**How to choose $\bar{d}$ and $\bar{m}$?** Although $\bar{d}$ and $\bar{m}$ are hyperparameters, it is worth noting that they can be determined in a highly principled way. In practice, given a search space, we only calculate the center $(\bar{d}^*, \bar{m}^*) = (\frac{\bar{d}_{\max}+\bar{d}_{\min}}{2}, \frac{\bar{m}_{\max}+\bar{m}_{\min}}{2})$ and the radius $(r_{\bar{d}}, r_{\bar{m}}) = (\frac{\bar{d}_{\max}-\bar{d}_{\min}}{2}, \frac{\bar{m}_{\max}-\bar{m}_{\min}}{2})$. We by default only keep architectures within half of the radius from the center for evaluation: $|\bar{d}-\bar{d}^*| \leq \frac{1}{2}r_{\bar{d}}$ and $|\bar{m} - \bar{m}^*| \leq \frac{1}{2}r_{\bar{m}}$. This principle leads to $(\bar{d}^*, \bar{m}^*) = (1.5, 10.5)$ with $\frac{1}{2}(r_{\bar{d}}, r_{\bar{m}}) = (0.7, 4.8)$ on DARTS, and $(\bar{d}^*, \bar{m}^*) = (1.6, 2.2)$ with $\frac{1}{2}(r_{\bar{d}}, r_{\bar{m}}) = (0.7, 0.9)$ on NAS-Bench-201.

**Implementation settings.** We train searched architectures for 250 epochs using SGD, with a learning rate as 0.5, a cosine scheduler, momentum as 0.9, weight decay as $3 \times 10^{-5}$, and a batch size as 768. This setting follows previous works [1, 44, 69, 41, 66, 26, 12, 10].

**DARTS space on ImageNet.** As shown in Table 1, for both two training-free NAS methods, by adopting our pre-filtration strategy, we can further reduce the search time cost and achieve better search results. For NAS-WOT, we can save 16.7% search time cost and reduce 0.3% top-1 error. For TE-NAS, we can significantly save 41.2% search time cost, and still improve the top-1 error by 0.4%. FLOPs of our search architectures are 0.68G (TE-NAS + DAG) and 0.56G (WOT + DAG).

**NAS-Bench-201 Space.** As shown in Table 2, for both NAS-WOT and TE-NAS, we reduce over 30% search time cost with strong accuracy. We also include a training-based method FP-NAS [65], where we even achieve 46% seach cost reduction with better performance. Moreover, we show that our method is robust to the choices of $\bar{d}$ and $\bar{m}$. In the ablation study in Table 3, by changing different ranges of $\bar{d}$ and $\bar{m}$, our method remains strong over the WOT baseline.

Table 3: Our method is robust to choices of $\bar{d}$ and $\bar{m}$ (WOT [44] on NAS-Bench-201).

| Ranges of accepted $\bar{d}$ and $\bar{m}$ | CIFAR-100 |
|---|---|
| $|\bar{d} - \bar{d}^*| \leq r_{\bar{d}}, |\bar{m} - \bar{m}^*| \leq r_{\bar{m}}$ (baseline) | 69.48 (1.70) |
| $|\bar{d} - \bar{d}^*| \leq \frac{3}{4}r_{\bar{d}}, |\bar{m} - \bar{m}^*| \leq \frac{3}{4}r_{\bar{m}}$ | 69.51 (1.70) |
| $|\bar{d} - \bar{d}^*| \leq \frac{1}{2}r_{\bar{d}}, |\bar{m} - \bar{m}^*| \leq \frac{1}{2}r_{\bar{m}}$ | 69.86 (1.24) |
| $|\bar{d} - \bar{d}^*| \leq \frac{1}{4}r_{\bar{d}}, |\bar{m} - \bar{m}^*| \leq \frac{1}{4}r_{\bar{m}}$ | 69.97 (1.19) |

## 6 Conclusion

In this work, we show that it is possible to conduct fine-grained convergence analysis on networks of complex connectivity patterns. By analyzing how an NNGP kernel propagates through the networks, we can fairly compare different networks' bounds of convergence rates. This theoretical analysis and comparison are empirically verified on MNIST and CIFAR-10. To make our convergence analysis more practical and general, we propose two intuitive principles on how to design a network's connectivity patterns: the effective depth and the effective width. Experiments on diverse architecture benchmarks and datasets demonstrate that networks with an extreme depth or width show bad performance, indicating that both the depth and width are important. Finally, we apply our principles to the large-scale neural architecture search application, and our method can largely accelerate the search cost of two training-free efficient NAS works with faster and better search performance. Our work bridge the gap between the Deep Learning theory and the application part, making the theoretical analysis more practical in architecture designs.

## Acknowledgement

B. Hanin and Z. Wang are supported by NSF Scale-MoDL (award numbers: 2133806, 2133861).

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
