# OpenReview forum: "Deep Architecture Connectivity Matters for Its Convergence: A Fine-Grained Analysis"
_NeurIPS.cc/2022/Conference — NeurIPS 2022 Accept_

### Official Review · Reviewer_6D9f · 2022-07-06

**Rating:** 7
**Confidence:** 4
**Soundness:** 3 good
**Presentation:** 3 good
**Contribution:** 3 good

**Summary:**

This paper studied how a wide network’s NNGP kernel can depict the optimization dynamics of a particular DNN topology, by propagating the NNGP kernel spectrum and showing the topology to affect the bound of convergence rate. Based on this observation, the authors created two notions of “efficient depth” and “effective width”, that can be plugged-in existing NAS methods to filter out “unpromising” connectivity patterns for speedup.

**Questions:**

See above

**Limitations:**

No particular negative impact. Limitation was discussed by authors.

**Strengths And Weaknesses:**

Strength:
This is an important new piece of work towards connecting deep learning theory and NAS. Prior arts already adapted theoretical properties of general DNNs, but never established their correlations with the concrete NN architecture topology, except some empirical correlation observations. This paper is the first to theoretically justify the optimization implication of general DNN topology, which is likely to become a milestone for future work in this frontier.
Throughout the paper, the theory and application aspects are tightly coupled, and the story is coherent. Their claim is theoretically sound (a non-surprising, yet nice adaptation of NNGP proofs). In the experiments, multiple benchmarks are reported, with error bars.

Section 5.3 is very helpful in understanding how the effective width/depth principles are used to improve NAS in a plug-in fashion. I especially like how the authors can choose d and m in a principled justified way, not ad-hoc.

Overall the writing is very good, clear and easy to follow. It’s a mature paper.

Weakness:
NNGP is a rough characterization of optimization dynamics, hence its preciseness in comparing architectures is limited. More importantly, as mentioned on lines 280, “Although our d and m are only inspired from the optimization perspective, not the complexity or generalization, our method mainly filters out bad architectures at a coarse level, but does not promote elites in a fine-grained way.” I was not sure whether this optimization-only selection bias will lead us to missing architectures that are excellent in complexity/generalization (hence offsetting their perhaps mediocre optimization behavior).
The authors applied their principles to accelerating TE-NAS and NAS-WOT. Those are two earliest training-free NAS approaches. Could the authors also try more recent ones and see improvements, e.g., Zen-NAS or zero-cost proxy NAS? Moreover, how about non-training-free NAS methods, can they be accelerated by such pre-filtering too?

In Tables 1 and 2, the FLOPS of searched architectures could be included too.

---

> ### Author Response · Authors · 2022-08-02
> **Response to Reviewer 6D9f**
>
> We sincerely thank the 6D9f's review.
>
> 1. **Limits of NNGP and convergence-based model selection.**
>
> Thanks for pointing this out! Quantifying the connection between the network’s topology with its complexity, generalization, and training dynamics are indeed important future works for us! Currently, our empirical results on diverse NAS methods and benchmarks show that: we can effectively compare different architectures, and improve both search accuracy and time cost by considering the network’s convergence (via measuring its effective depth and width).
>
> 2. **Non-training-free NAS methods.**
>
> To make our experiments stronger, we further study a training-based NAS method, FP-NAS [1]. We will include this table in our camera ready. We can reduce 46% the search time cost, and boost its performance on all three datasets on NAS-Bench-201 (with standard deviations over three random runs in parentheses).
> | Method | Search (s) | CIFAR-10 | CIFAR-100 | ImageNet-16-120 |
> |:---:|:---:|:---:|:---:|:---:|
> | FP-NAS [Yan 2021] | 4837 | 77.4 (16.6) | 64.7 (5.3) | 26.7 (10.4) |
> | FP-NAS + DAG (ours) | 2612 (-46.0\%) | 93.3 (0.3) | 70.8 (0.4) | 44.5 (1.4) |
>
> 3. **FLOPS of searched architectures.**
>
> FLOPs of our search architectures on ImageNet are 0.68G (TE-NAS + DAG) and 0.56G (WOT + DAG); on CIFAR-10 on NAS-Bench-201, FLOPs are 0.18G (TE-NAS + DAG) and 0.15G (WOT + DAG). We will include them in our camera ready. Meanwhile, previous NAS works we compared with did not report their FLOPs.
>
> [1] Yan et al. “FP-NAS: Fast Probabilistic Neural Architecture Search.” *CVPR 2021*

---

> > ### Comment · Reviewer_6D9f · 2022-08-08
> > **Responses to the reply**
> >
> > Thanks for the responses from the authors. I agree with the authors' response and the rate will not change.

---

> > > ### Author Response · Authors · 2022-08-08
> > > **Thanks for your further response!**
> > >
> > > We appreciate your time and response! We will include your suggestions in our camera ready!

---

### Official Review · Reviewer_D7qw · 2022-07-11

**Rating:** 7
**Confidence:** 4
**Soundness:** 4 excellent
**Presentation:** 3 good
**Contribution:** 3 good

**Summary:**

The authors proposed a formal analysis framework to estimate the the upper bound of training loss convergence for DAGs with variety of network topologies. Based on that, they proposed a plug-and-play method to speech up previously reported NAS methods by apply a filtration. The results demonstrate the proposed method to search better networks and faster.

**Questions:**

See above.

**Limitations:**

Current discussion is sufficient.

**Strengths And Weaknesses:**

+ To my best knowledge, this is perhaps the first theoretical study directly focusing on the fine-grained NN topological connectivity rather than a general NN function, despite some prior works exploring part of that, e.g., [50] for NN width/depth, [69] for skip connection. It is an important yet understudied direction, and this work appears to lay a good foundation.

+ I have gone over the details of the derivation for the convergence analysis of DNN regarding the connectivity patterns, and it seems sound. For the bound estimation, the authors chose a simplified theoretical model of DAGs + MSE loss, using NNGP kernel. The key step is to estimate the flow of NNGP variance and mean through unidirectional information paths in the specific topology. The theoretical results are examined by a series of simulations in Sec 5.1 and Sec 5.2.

+ The notions of ‘effective depth’ and ‘effective width’ are clearly defined, and the authors also gave clear guidelines how to use them in NAS. It is shown to accelerate two latest NAS methods and outperform more, across multiple benchmarks. The experiments validate the effectiveness of the proposed method.

In general, I think this is a cool and well-polished paper. One question is the authors demonstrated their down-selection technique to two training-free NAS approaches (NAS-WOT and TE-NAS), which are already very fast and therefore show only non-significant reduction of search time. I wonder why not trying the proposed approach on more costly and accurate search methods, and see if the accuracy-search efficiency gain is still favorable.

---

> ### Author Response · Authors · 2022-08-02
> **Response to Reviewer D7qw**
>
> We sincerely thank the D7qw's review.
>
> **Question: Why not try on more costly search methods?**
>
> **Answer:** To make our experiments stronger, we further study a training-based NAS method, FP-NAS [1]. We will include this table in our camera ready. We can reduce 46% search time cost, and boost its performance on all three datasets on NAS-Bench-201 (with standard deviations over three random runs in parentheses).
> | Method | Search (s) | CIFAR-10 | CIFAR-100 | ImageNet-16-120 |
> |:---:|:---:|:---:|:---:|:---:|
> | FP-NAS [Yan 2021] | 4837 | 77.4 (16.6) | 64.7 (5.3) | 26.7 (10.4) |
> | FP-NAS + DAG (ours) | 2612 (-46.0\%) | 93.3 (0.3) | 70.8 (0.4) | 44.5 (1.4) |
>
>
> [1] Yan et al. “FP-NAS: Fast Probabilistic Neural Architecture Search.” *CVPR 2021*

---

### Official Review · Reviewer_rAbP · 2022-07-11

**Rating:** 6
**Confidence:** 4
**Soundness:** 3 good
**Presentation:** 4 excellent
**Contribution:** 3 good

**Summary:**

The paper theoretically and empirically analyzes the link between connectivity patterns of deep networks and their convergence. Based on this analysis, the authors propose a couple of training-free metrics (effective width and effective depth) and use them to do training-free NAS. The paper presents a thorough link between gradient descent convergence and structural properties of deep networks by proving a link between number of unique paths, number of incoming paths at certain nodes in the network and the least eigenvalue of a Neural Network Gaussian Process (NNGP) variance. Essentially, the authors look at how the NNGP variance changes as the data propagates through the model and relate it to how many unique paths it goes through and how many incoming edges land at the last layer (in the DAG topology used to formulate this problem). These principles are then baked into their “effective depth” and “effective width” metrics that do not require any training or forward/backward passes to compute. Some empirical results are shown for this convergence. The remaining paper looks at training-free NAS by cutting down the search space. Results are shown on CIFAR-10/100, ImageNet-16-120, and ImageNet.

**Questions:**

Please see the weaknesses section above (particularly points 2 and 3).

**Limitations:**

Authors could comment on role of kernel sizes, etc., and how the connectivity alone does not take this into account in the present study.

**Strengths And Weaknesses:**

The paper has following strengths:

1. This is a principled approach to study an important problem, i.e., understanding how deep network architecture topologies and connectivity patterns relate to training convergence. Knowing this is obviously very valuable because (a) we can design better models directly without a long, time-consuming search, (b) in future, better understanding of how architectures impact accuracy can enable us to create even more novel models. Indeed, it is a hard problem, so coming up with an intuitive method for this problem is very hard.

2. The proof seems very intuitive and the resulting metrics (effective depth and effective width) are very simple to compute and do not require any training or even forward/backward passes. One can just look at the architecture connectivity and tell with reasonable confidence if it is a good or bad architecture.

3. Empirical results for training convergence and NAS-bench architectures, etc., present some interesting insights (although, they also bring up some questions, see below).

4. Authors have shown experiments all the way to ImageNet which is nice.

Despite the strengths, there are some weaknesses and questions. Addressing these would likely make the paper much stronger:

1. The empirical validation of the theory (section 5.1) is done using only three architecture topologies. This is not that expensive (particularly for MNIST and CIFAR-10). Can the authors look at many different deep nets with different connectivity patterns and somehow present more data on convergence?

2. Experiments section could in general be stronger. For instance, the ImageNet result on TE-NAS is not good enough. We see some accuracy improvement, but we also see model size increase compared to vanilla TE-NAS. Moreover, since the training-free NAS itself is very cheap, further improvements in search time do not make a significant difference. In Figure 5, each point represents a subset of models that achieved a similar accuracy. Were there any interesting patterns in terms of their number of parameters/MACs? Example: did the white/black circles contain models of a certain size in terms of parameter counts? Were the models deeper but narrower within a circle? Or shallower but wider?

3. The very first paper that pioneered the creation of such connectivity-based metrics was reference [R1] cited by the authors. This paper needs to be discussed in much more detail in Section 2.2/2.3 as there are many interesting synergies. For instance, [R1] showed the link between connectivity patterns and training convergence too. [R1] showed very concrete training convergence curves and showed that their proposed metric correlated very well with convergence. The metric proposed by [R1] is also training-free. Indeed, the present study is much more fine-grained, more theoretical, and is more generally applicable than [R1].

4. Other weaknesses involve things like there is no characterization of differences in kernel sizes, etc., but this is not critical and can be a good future work.

5. Proof for equation (14) in appendix B has a typo? K_{ii}^{l} should have a square on \sigma (right below “Proof of Lemma 3.2”)? Honestly, this paper could be much stronger if there was a bit more focus on empirical results for justifying the theory (e.g., if section 5.1 was much stronger).

[R1] Bhardwaj, Kartikeya, Guihong Li, and Radu Marculescu. "How does topology influence gradient propagation and model performance of deep networks with DenseNet-type skip connections?." Proceedings of the IEEE/CVF Conference on Computer Vision and Pattern Recognition. 2021.

---

> ### Author Response · Authors · 2022-08-02
> **Response to Reviewer rAbP**
>
> We sincerely thank the rAbP’s review.
>
> 1. **More connectivity patterns and convergence.**
>
> We sample 408 networks from our DAG space, and train them all on CIFAR-10. We provide a comprehensive study on architectures’ topologies with their empirical convergence, and include it in Appendix G in our updated draft. Again, we find networks with moderate effective depth and width show faster convergence.
>
> 2. **Stronger experiments.**
>
> To make our experiments stronger, we further study a training-based NAS method, FP-NAS [1]. We will include this table in our camera ready. We can reduce 46% search time cost, and boost its performance on all three datasets on NAS-Bench-201 (with standard deviations over three random runs in parentheses).
> | Method | Search (s) | CIFAR-10 | CIFAR-100 | ImageNet-16-120 |
> |:---:|:---:|:---:|:---:|:---:|
> | FP-NAS [Yan 2021] | 4837 | 77.4 (16.6) | 64.7 (5.3) | 26.7 (10.4) |
> | FP-NAS + DAG (ours) | 2612 (-46.0\%) | 93.3 (0.3) | 70.8 (0.4) | 44.5 (1.4) |
>
> 3. **Number of parameters/MACs in Figure 5.**
>
> In our Figure 5, for a fair comparison, we fixed the number of parameterized layers in all architectures to be the same (three parameterized layers). They have FLOPs of 113.95 M and parameters of 0.802 MB. Architectures in each circle share the same effective depth and width, with possibly different fine-grained topologies (connectivities).
>
> 4. **More discussions on Bhardwaj et al. 2021.**
>
> Thanks for pointing out this important work! Comparing the two works, the core difference is that, we start from the analysis of the deep network’s convergence rate, and then link the convergence to the network’s topology. At the same time, Bhardwaj et al. proposed NN-Mass by analyzing the network’s layer-wise dynamic isometry, and then it is empirically linked to the network’s convergence. Both two approaches are insightful and interesting, and we will make sure to include more discussion in our camera ready.
>
> 5. **Kernel sizes.**
>
> The size of the NNGP kernel is #samples$\times$#samples. Larger kernels will have smaller $\lambda_\min$.
>
> 6. **Typo in Eq. 14.**
>
> Yes, it is a typo. Thank you and we have fixed it!
>
>
> [1] Yan et al. “FP-NAS: Fast Probabilistic Neural Architecture Search.” *CVPR 2021*

---

> > ### Comment · Reviewer_rAbP · 2022-08-08
> > **Thanks for the response**
> >
> > I have read author response and other reviews. I will maintain my rating of 6. While it is good that new experiments generally agree with authors' claim "moderate effective depths and effective widths work well and neither extreme effective width nor extreme effective depth work", the new results in Appendix G show that models with effective width of 4 (which is its maximum value) also reach convergence quickly. This means that there is still gap in the complete understanding of such methods and, like prior theoretically grounded topological metrics (e.g., Bhardwaj et al. 2021), there should be upper and lower limits at which effective width and effective depth stop working. Thus, better analysis and understanding can improve this method. Another weakness is that in Fig. 4, eventually the slower convergence network outperforms the fast convergence network in the end. If there is a way to understand this phenomenon, that would be even better. However, improving upon the current work should definitely be a future work. It is clear that this line of thinking (i.e., connectivity pattern analysis) is producing significantly cheaper methods to design promising models. My recommendation is to accept this paper in its current form.

---

> > > ### Author Response · Authors · 2022-08-08
> > > **Thanks for your further comments!**
> > >
> > > We greatly appreciate your further comments!
> > >
> > > We agree that convergence is a part of the whole picture of a network's property. Especially, a network's generalization and functional complexity are also important aspects.
> > > Quantifying the connection between the network’s topology with its complexity, generalization, and training dynamics are indeed important future works for us!

---

### Author Response · Authors · 2022-08-02
**We thank all reviewers for recognizing our work!**

We truly thank all questions and suggestions from three reviewers.

We are very happy to see all reviewers acknowledge that:

1) We study an important problem of understanding fine-grained NN topological connectivity, and lay a good foundation and a milestone for future work.

2) Our method is principled, intuitive, and theoretically sound.

3) Our metrics (effective depth and width) are clearly defined, simple to compute, chosen in a principled way, with clear usage guidelines in NAS and a tightly coupled theory-application story.

4) Our paper writing is well-polished and easy to follow.

We address all questions below.

---

### Meta-Review · Area_Chair_gVu2 · 2022-08-27

**Recommendation:** Accept
**Confidence:** Certain

**Metareview:**

This paper studies the relationship between connectivity of a deep network and its convergence, both theoretically and empirically. The paper also studies simpler metrics such as effective depth and width to guide the architecture search. Overall this is an impressive theoretical paper supported by empirical evidences.

All the three reviewers find the paper a valuable contribution to an important theoretical problem in deep learning. After reading the rebuttals, Reviewer rAbP recommended to accept this paper in its current form. Reviewer D7qw felt that all the concerns had been well addressed, and increased the score by one. Reviewer 6D9f agreed with the authors' response.

**Award:**

No

---

### Decision · Program_Chairs · 2022-09-14

Accept